# Microalga Biofertilizer Triggers Metabolic Changes Improving Onion Growth and Yield

**Ely Cristina Negrelli Cordeiro, Átila Francisco Mógor \*, Juliana de Oliveira Amatussi, Gilda Mógor, Gabriel Bocchetti de Lara and Harielly Marianne Costa Marques**

Crop Science Department, Federal University of Paraná, Curitiba 80035-050, Brazil; elycordeiro@ufpr.br (E.C.N.C.); juliver@ufpr.br (J.d.O.A.); gilda.mogor@ufpr.br (G.M.); gabriel.lara@ufpr.br (G.B.d.L.); hariellymarques@ufpr.br (H.M.C.M.)
\* Correspondence: atila.mogor@ufpr.br

**Abstract:** Seeking the development of nature-friendly agronomic techniques, the use of natural sources to promote plant growth and increase agricultural yield has gained relevance. In this context, the use of biofertilizers or biostimulants obtained from microalgae has been studied, as these microorganisms have in their composition a great diversity of bioactive molecules. This study aimed to evaluate the effect of microalga *Asterarcys quadricellulare* (CCAP 294/1) on organic onion production, verifying its action on metabolism, growth and yield of two cultivars. Thus, two experiments were carried out: (i) foliar applications on onion plants grown in pots in a greenhouse; (ii) foliar applications on field-grown onion under an organic system. Both experiments were undertaken using solutions with spray-dried microalga biomass at concentrations of 0.05, 0.15, 0.25 and 0.4 g L$^{-1}$. Biometric variables, yield of bulbs and biochemical variables were evaluated indicating that the use of *A. quadricellulare* promoted plant growth and increases in bulb caliber and yield of both onion cultivars. The microalga biomass stimulated plant metabolism by increases in contents of chlorophyll, carotenoids, amino acids, and the nitrate reductase enzyme activity in leaves, also free amino acids and total sugar contents in bulbs, highlighting the biomass concentration of 0.25 g L$^{-1}$.

**Keywords:** *Allium cepa* L.; *Asterarcys quadricellulare*; biostimulant

## 1. Introduction

Organic agriculture uses methods that minimize damage to the environment, producing food sustainably [1]. In this sense, the use of biofertilizers has expanded its applicability and may contribute to an increase in yield [2,3].

Recently, numerous species of microalgae have been studied for their plant growth-promoting effect linked to their use as biofertilizers and biostimulants [4] especially associated with several bioactive compounds present in these organisms, such as polysaccharides, glycosides, phytohormones, polyamines, lipids, and free L-amino acids [3,5–9].

As examples, the use of the green microalgae (Chlorophyta) *Chlorella vulgaris* promoted growth in corn and wheat plants [10,11]. In lettuce, *C. vulgaris* increased protein content and leaf mass [12], and higher onion bulb caliber and yield were obtained using the Chlorophyta *Scenedesmus subspicatus* [4]. In tomatoes, the use of *Acutodesmus domorphus* applied to seedlings promoted biomass accumulation and flowering [13]. In addition, the bioactivity of a biomass hydrolysate from the cyanobacteria *Asthrospira platensis* in bioassays was identified, promoting growth and production of lettuce [6]. The use of *A. platensis* also showed bioactivity in bioassays and increased sugar beet production in organic systems by using foliar sprays [7].

The microalgae Clorophyta *Asterarcys quadricellulare* (AQ) is a source for the production of carotenoids [14] and shows high protein and carbohydrate contents [15], therefore, presenting potential for application as a biofertilizer.

Onion (*Allium cepa*) is the third most produced and consumed vegetable in the world, considered as of great economic importance. Brazil is the 12th largest onion producer worldwide, with 1,550,000 tons in 2018 [16].

In this context, this work aimed to evaluate the use of the microalgae *A. quadricellulare* biomass verifying its action on metabolism, growth and yield of two onion cultivars.

## 2. Materials and Methods

The following experiments were performed: (i) foliar sprays of *A. quadricellulare* spray-dried biomass on onion plants grown in pots in a greenhouse determining biochemical alterations and biometric variables of growth; (ii) foliar sprays on organic field-grown onion plants; determining classification and yield of bulbs, as well as biochemical variables in leaves (chlorophyll, carotenoids, amino acids and the activity of the enzyme nitrate reductase) and in bulbs (free amino acids and total sugars).

### 2.1. Microalga Source and Treatments

The microalga *Asterarcys quadricellulare* (CCAP 294/1) biomass from Alltech Crop Sciences, Brazil, is a fine greenish powder obtained by spray drying method from mixotrophic cultivation. The L-free amino acids concentration in the biomass was of 90.94 mg g$^{-1}$, equivalent to 9% in mass, determined using 0.1 mg of dry biomass diluted in 1.7 ml of 80% ethanol preparing an extract from which 1.0 mL was diluted in distilled water [17]. The colorimetric reaction was also carried out [18]. The biomass was diluted in suspensions containing the following concentrations: 50, 150, 250 and 400 g L$^{-1}$, from which the 1 mL aliquot was removed, diluted in one liter of distilled water, giving rise to treatment equivalent to concentrations of 0.05 g L$^{-1}$ (AQ 5), 0.15 g L$^{-1}$ (AQ 15), 0.25 g L$^{-1}$ (AQ 25), and 0.4 g L$^{-1}$ (AQ 40).

### 2.2. Study Area, Cultivars, and Seedling Production

The experiment was performed in an organic system at the Experimental Station on Canguiri farm of the Federal University of Paraná UFPR, at latitude 25°23′30″ south and longitude 49°07′30″ west, average altitude of 920 m, state of Paraná, Brazil. The region's climate is temperate, humid, mesothermal, cfb-type, according to Koppen's classification.

Two onion cultivars were used for the experiment. The cultivar Alvará (Bejo®), with early maturation, high productivity, vigorous leaves growth, with good waxiness, gives it great versatility in terms of planting time. Its cycle lasts from 130 to 150 days and its sowing is recommended between May and June. The hybrid cultivar Perfecta (Topseed®) has high yield potential, good tolerance to diseases and excellent bulb quality, vigorous leaves, good waxy and dark green colored leaves. It is a short-day hybrid, with good tolerance to early bolting. Its cycle is 130–150 days, with sowing time indicated for the southern region from April to July.

In mid-May 2018, cultivars were sown in beds under plastic-covered high type tunnels. At 60 days after sowing (DAS), when transplanting was carried out, the seedlings had five leaves and pseudostem diameters of 4 mm.

### 2.3. Pot Experiment

The onion seedlings were transplanted into 3 L polyethylene pots containing a substrate on the basis of pine bark, peat, expanded vermiculite, enriched with macro and micronutrients (Tropstrato®) combined with organic fertilizer (Provaso®), with a proportion of 1:1. The substrate presented the following chemical analysis: pH (CaCl$_2$) = 6.63; pH SMP = 7.03; Al$^{3+}$ = 0; H$^+$ + Al$^{3+}$ = 2.31 cmol dm$^{-3}$; Ca$^{2+}$ = 12.34 cmol dm$^{-3}$; Mg$^{2+}$ = 3.52 cmol dm$^{-3}$; K$^+$ = 1.95 cmol dm$^{-3}$; P = 193.81 mg dm$^{-3}$; C = 55.21 g dm$^{-3}$; base saturation = 88.5% and CEC = 20.12 cmol dm$^{-3}$; Cu = 1.95 mg kg$^{-1}$; Mn = 31.80 mg kg$^{-1}$; Fe = 29.77 mg kg$^{-1}$; Zn = 2.30 mg kg$^{-1}$; B = 0.34 mg kg$^{-1}$; S = 141.12 mg kg$^{-1}$. Additionally, 3 cm$^3$ of expanded vermiculite was added to the top of the pots. The pots were placed in

a greenhouse, with drip irrigation to maintain humidity at 80% of the substrate's water holding capacity, measured with a tensiometer.

The biomass of *Asterarcys quadricellulare* (AQ) was suspended at the following concentrations: 50, 150, 250, and 400 g $L^{-1}$. An aliquot of 1 mL $L^{-1}$ was withdrawn from each suspension and diluted in distilled water, resulting in solutions with biomass concentrations of 0.05 g $L^{-1}$ (AQ 5), 0.15 g $L^{-1}$ (AQ 15), 0.25 g $L^{-1}$ (AQ 25), and 0.4 g $L^{-1}$ (AQ 40).

The foliar sprays started at 40 days after transplanting (DAT), with a weekly interval, totaling eight applications until the plants were collected, using a Kawashima® (Tokyo, Japan) electronic sprayer at pressure of 40 psi. The sprays volume varied from 5 mL plant$^{-1}$ (1st to 3rd applications) to 10 mL plant$^{-1}$ (4th to 6th applications) and 15 mL plant$^{-1}$ (7th and 8th applications), with increments following the plant growth.

At 90 DAT, to check the AQ effect on biomass accumulation and plant growth at the beginning of bulbification, all plants of both cultivars were collected for biometric evaluations of fresh mass and length of leaves, pseudostem diameter, and determinations of chlorophyll, sugars, amino acids and nitrate reductase enzyme activity in leaves.

With completely randomized design and factorial scheme, the experiment was run with four replications ($n = 4$), each consisting of four pots containing two plants per pot.

### 2.4. Field Experiment

Simultaneously to the experiment in greenhouse, the field experiment was conducted in the organic vegetables area described above, in a Red-Yellow latosol soil of medium texture [19].

The chemical analysis of the soil at the beginning of experiment in the 0–20 cm layer indicated the mean values of: pH ($CaCl_2$) = 5.84; pH $H_2O$ = 6.71; $Al^{3+}$ = 0; H + $Al^{3+}$ = 2.93 cmolc dm$^{-3}$; $Ca^{2+}$ = 5.28 cmolc dm$^{-3}$; $Mg^{2+}$ = 3.05 cmolc dm$^{-3}$; $K^+$ = 1.32 cmolc dm$^{-3}$; P (Mehlich) = 49.0 mg dm; S = 33.49 mg dm$^{-3}$; C = 26 g dm$^{-3}$; %; V% = 76.7 and CEC = 12.58 cmolc dm$^{-3}$. Sprinkler irrigation was used to maintain the soil moisture at 80% and checked with a tensiometer.

One week before seedling transplantation, soil tillage was performed with the incorporation of 8 t.ha$^{-1}$ of organic compost with the following average values: C = 30.3 g kg$^{-1}$; N = 30.3 g kg$^{-1}$; P = 8.5 g kg$^{-1}$; K = 6.6 g kg$^{-1}$; Ca = 8.1 g kg$^{-1}$; Mg = 4.1 g kg$^{-1}$, following the Brazilian regulations for organic farming.

The onion seedlings of Alvará and Perfecta cultivars were transplanted in beds measuring 1.20 × 24 m, with a spacing of 30 cm between rows and 10 cm between plants, distributed in four planting rows, equivalent to a plant population of 240,000 per hectare.

The treatments consisted of foliar sprays of microalgae suspensions as described in the pot experiment. Ten applications were carried out using a 10 L Kawashima® electronic backpack sprayer, with an application rate of 400 L ha$^{-1}$, at intervals of 7 days. The experiment was established in a completely randomized design in a factorial scheme (A: two cultivars x B: 5 treatments, with 4 replications ($n = 4$)), totaling 40 plots with 60 plants (1.20 × 1.50 m).

At 120 DAT, leaves were collected for biochemical analysis at around 9 a.m. and 10 a.m., with five onion plants randomly chosen per plot and the four central leaves of these plants were collected.

At 135 DAT, at proper harvest time with about 80% of the plants presenting pseudostem collapse, 20 bulbs per plot were collected to determine the yield in commercial classes: class IV (70–90 mm) and class III (50–70 mm) according to the Brazilian market rules for onion classification, extrapolating to the population of plants per hectare [4]. The presentation of class data was based on the mass of bulbs (g) shown in classes III and IV.

The fresh mass and dry mass of bulbs were also determined in an air-circulating oven at 65 °C until a constant weight was obtained. To determine the biochemical variables of the bulbs, five onion plants were randomly chosen per plot, and the bulbs of these plants were evaluated.

*2.5. Biochemical Analysis*

After collection, the plant materials were frozen and subsequently macerated. The readings were completed using a spectrophotometer with values expressed in $\mu g\ g^{-1}$ of the fresh mass of leaves and bulbs.

The determination of chlorophyll and carotenoids was carried out following [20] with modifications [21]. For extraction of total and reducing sugars from bulbs and leaves [22], samples were obtained by performing acid hydrolysis and after reacting with DNS. The readings were carried out at 540 nm.

The extraction of total free amino acids from leaves and bulbs [17] and colorimetric reaction [18] were performed, with readings carried out at 570 nm. The nitrate reductase enzyme activity was determined in leaves [23], with readings carried out at 540 nm and values expressed in $\mu$ mol of $NO_2\ h^{-1}\ g^{-1}$ of plant material.

*2.6. Statistical Treatment*

The data were subjected to ANOVA in a $2 \times 5$ factorial arrangement (cultivars x treatments) and compared by Tukey's test at 5% significance, using the software Assistat 7.7 Beta [24].

**3. Results**

*3.1. Pot Experiment*

The fresh mass of leaves presented increases, promoted by AQ 5 and AQ 15 in Alvará of 34.65 and 48.75% over the control, respectively; as well as a remarkable increase was promoted by AQ 5 and AQ 25 in Perfecta, respectively, of 151.87 and 162.73% over the control (Figure 1a). The length of leaves was improved in 18% by AQ 25 at Alvará, and in 20.23, 19.36, and 23.66% by AQ 15, AQ 25, and AQ 40, respectively, in Perfecta (Figure 1b).

The AQ 15, AQ 25, and AQ 40 increased the pseudostem diameter, respectively, of 23.53, 11.9, and 13.85% in Alvará; while AQ 5; AQ 25 and AQ 40 increased the diameter respectively of 23.3; 25.76 and 23.15% in Perfecta.

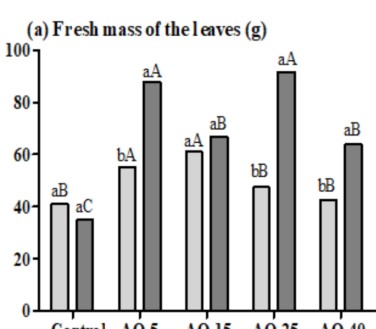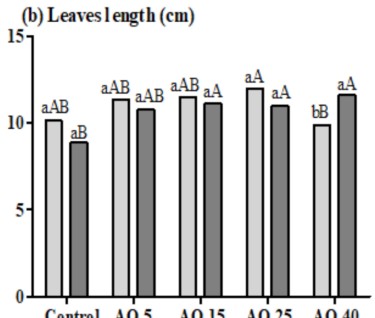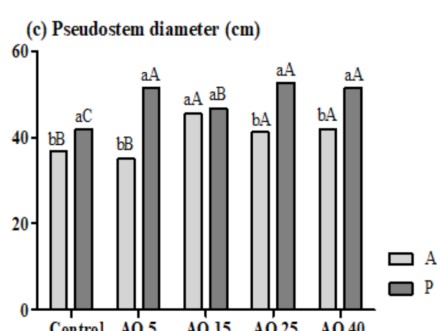

**Figure 1.** Fresh mass of the leaves (**a**), leaf lengths (**b**), pseudostem diameter (**c**) of onion plants of the Alvará and Perfecta cultivars at 90 DAT (days after transplanting) submitted to foliar sprays of biomass suspensions of microalgae *A. quadricellulare* at different concentrations. Treatments: $0\ g\ L^{-1}$ (control); $0.05\ g\ L^{-1}$ (AQ 5); $0.15\ g\ L^{-1}$ (AQ 15), $0.25\ g\ L^{-1}$ (AQ 25); $0.4\ g\ L^{-1}$ (AQ 40). Onion cultivars (A = Alvará, P = Perfecta). Lower-case letters in the column: cultivars; upper-case letters in the column: treatments. Means followed by the same letter in the same column do not differ statistically ($p < 0.05$) according to Tukey's test ($n = 4$).

In addition to the effects on growth (Figure 1), biochemical changes also occurred. Comparing cultivars, Perfecta had higher chlorophyll content in comparison to Alvará (27%). Comparing treatments, AQ 25 excelled in the cultivar Alvará with 44.71% greater chlorophyll content than the control. AQ 5, AQ 15, AQ 40 in the cultivar Perfecta improved chlorophyll to 34.1, 31.75 and 42.91% (Figure 2a).

For carotenoids, Perfecta presented a concentration 25.5% higher than Alvará. Comparing treatments, AQ 25 showed improved carotenoids content in 59.15% higher than the

control in Alvará, and in Perfecta, AQ 5 and AQ 40 promoted, respectively, increases of 19.06 and 27.37% (Figure 2b).

As for the total free amino acid content in leaves, Perfecta showed a concentration 18.92% higher than Alvará. Comparing treatments, AQ 15 promoted an increase of 38.85% in Alvará, while AQ 5, AQ 15, and AQ 40 improved free amino acids of 17.41, 19.89, and 17.08% above the control, respectively, in Perfecta (Figure 2c). For the activity of the nitrate reductase enzyme, no interaction was found among treatments and cultivars. Whereas, there was an effect of treatments, with higher enzyme activity promoted by AQ 25 and AQ 40 in comparison to the control (18 and 23.16%) (Figure 2d).

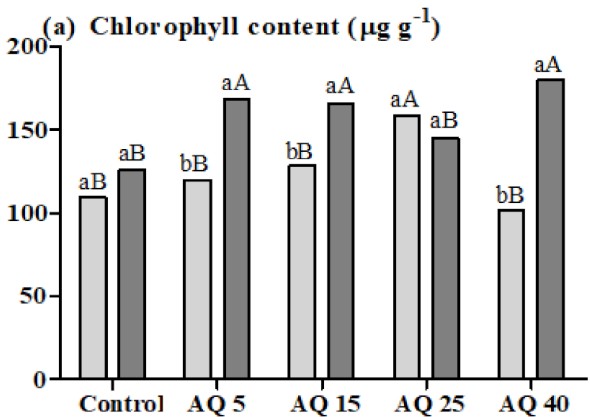
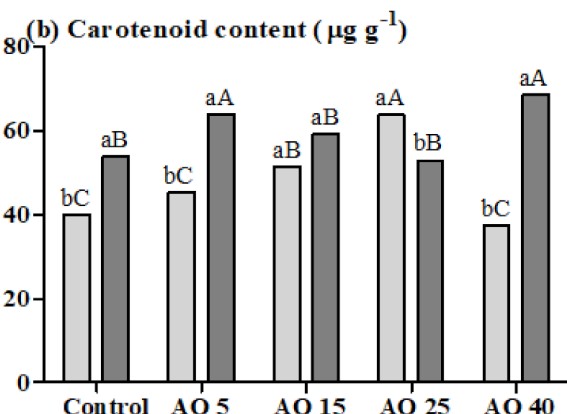
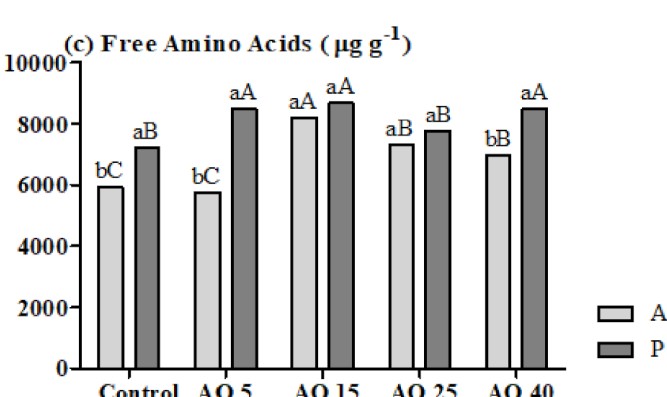
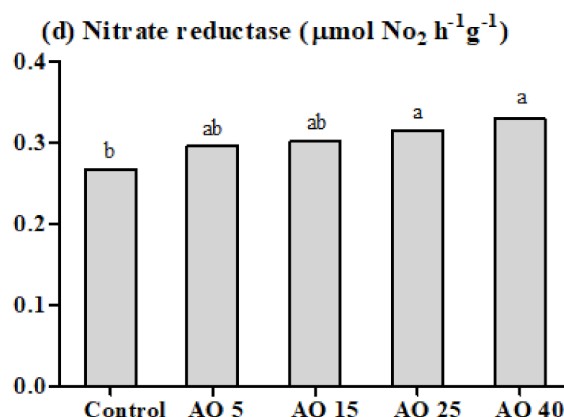

**Figure 2.** Chlorophyll content (**a**), carotenoid content (**b**), free amino acids, (**c**) nitrate reductase (**d**) in leaves of onion cultivars submitted to foliar applications of biomass suspensions of the microalgae A. quadricellulare at different concentrations. Treatments: $0 \, \text{g L}^{-1}$ (control); $0.05 \, \text{g L}^{-1}$ (AQ 5); $0.15 \, \text{g L}^{-1}$ (AQ 15), $0.25 \, \text{g L}^{-1}$ (AQ 25); $0.4 \, \text{g L}^{-1}$ (AQ 40). Onion cultivars (A = Alvará; P = Perfecta). Lower-case letters in the column: cultivars; upper-case letters in the column: treatments. Averages followed by the same letter in the same column do not differ statistically among themselves ($p < 0.05$) according to Tukey's test ($n = 4$).

### 3.2. Field Experiment

The data of commercial onion bulbs, class IV (70–90 mm) and class III (50–70 mm), showed in Alvará an interaction among classes and treatments, and a difference between treatments was observed in class IV, particularly at AQ 25 with an increase of 30.19% in this class when compared to the control. In class III, the highest means are found in treatments AQ 15 and AQ 25, with an increase of 16.83 and 11.73%.

The Perfecta showed no interaction among classes and treatments, but all treatments promoted increments in commercial bulbs in comparison to the control (Figure 3b).

Bulb fresh mass did not show an interaction among cultivar and treatment, but differences between treatments were found. The highest means of fresh mass of bulbs were observed in treatments AQ 5, AQ 15, and AQ 25, with an increase of 26, 22.56, and 24.79%, respectively, comparing to the control (Figure 3c). The same was observed for dry mass. The treatments that showed the highest means were AQ 5, AQ 15, and AQ 25, with an increase of 42.42, 49.43, and 47.5% when compared to the control dry mass (Figure 3d).

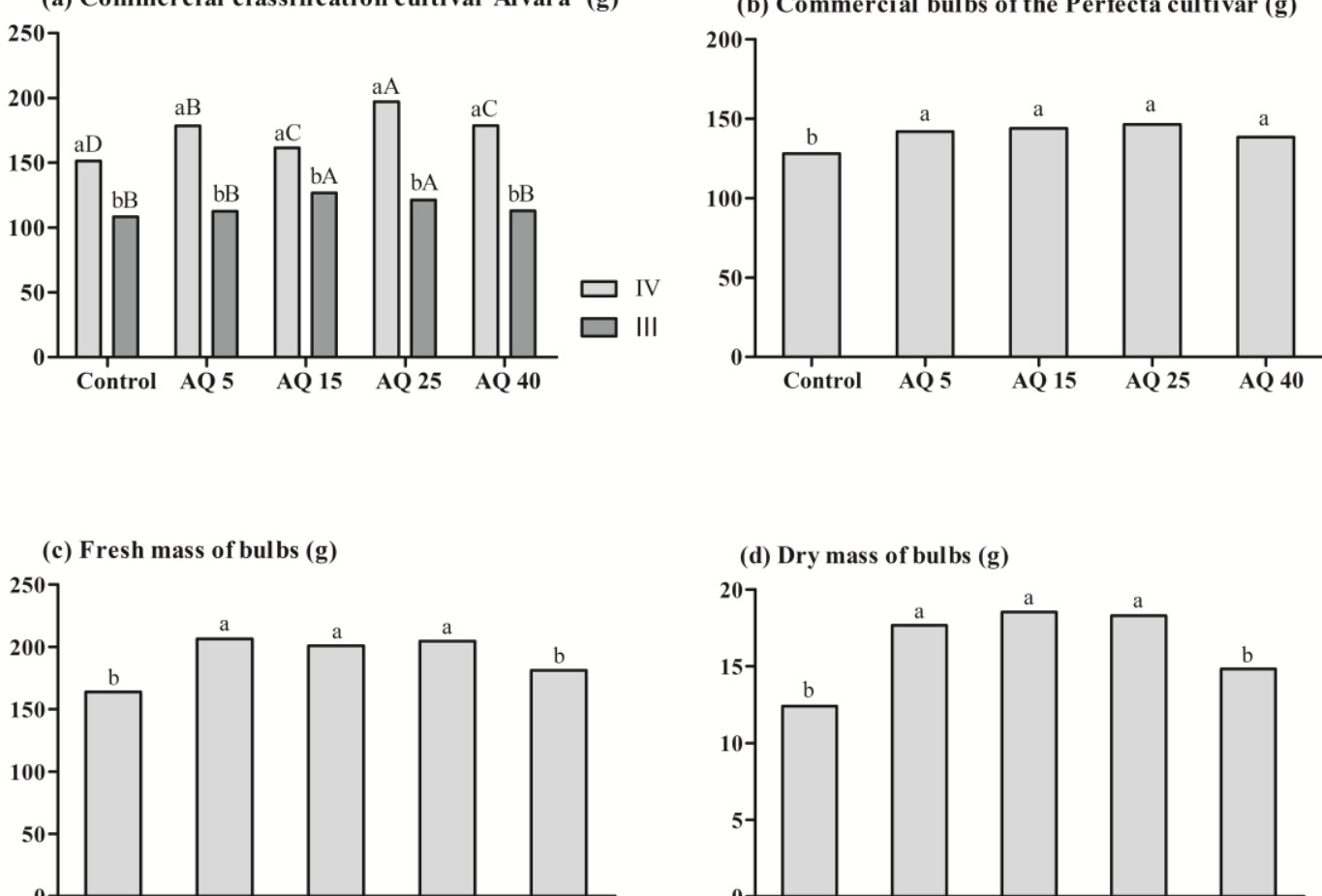

**Figure 3.** Classification of class III and IV bulbs of the Alvará onion cultivar (**a**), average classification of class III and IV bulbs of cultivar Perfecta (**b**), fresh mass of bulbs of the Alvará and Perfecta cultivars (**c**), dry mass of bulbs of the Alvará and Perfecta cultivars (**d**) submitted to foliar applications of biomass suspensions of the microalgae A. quadricellulare at different concentrations. Treatments: 0 g L$^{-1}$ (control); 0.05 g L$^{-1}$ (AQ 5); 0.15 g L$^{-1}$ (AQ 15), 0.25 g L$^{-1}$ (AQ 25); 0.4 g L$^{-1}$ (AQ 40). Class = IV and III. Columns with the same letter do not differ statistically ($p < 0.05$) by Tukey's test ($n = 4$). Capital letters = foliar treatments. Lower-case letters = classes.

As a consequence of the changes in the classification and masses of the bulbs, there were increases in onion yield with the application of the AQ treatments.

In Alvará, AQ25 promoted yield increase of 28.3%, while in Perfecta, the treatments AQ 5, AQ 15, and AQ 25 promoted yield increases by 26.36, 33.26 and 39.97%, respectively (Figure 4). These results demonstrate the effect of *A. quadricellulare* promoting higher masses of commercial bulbs.

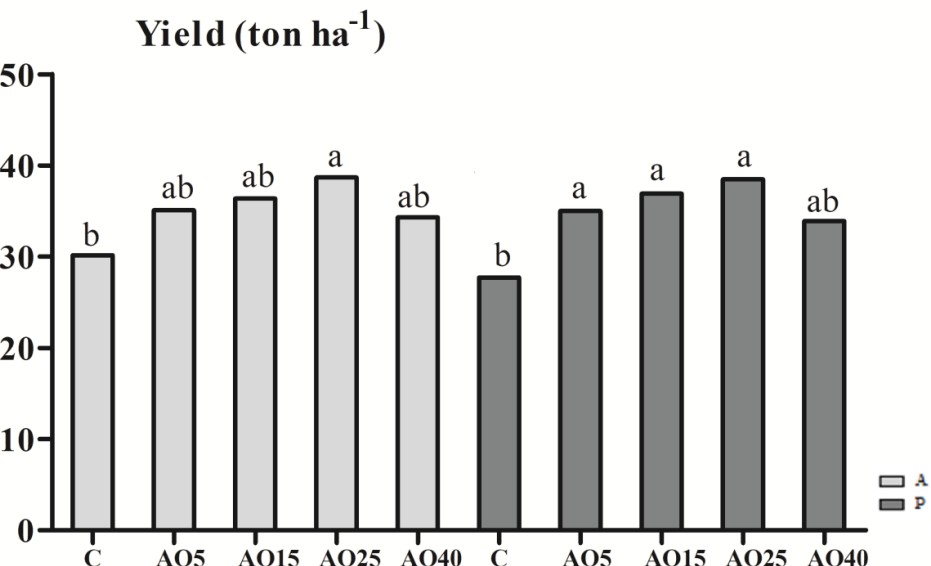

**Figure 4.** Yield of Perfecta and Alvará onion cultivars submitted to foliar applications of A. quadricellulare microalgae biomass suspensions at different concentrations. Treatments: 0 g L$^{-1}$ (control); 0.05 g L$^{-1}$ (AQ 5); 0.15 g L$^{-1}$ (AQ 15); 0.25 g L$^{-1}$ (AQ 25); 0.4 g L$^{-1}$(AQ 40). Means followed by the same letter in the same column do not differ statistically ($p < 0.05$) according to Tukey's test ($n = 4$).

The yield gains are related to biochemical changes stimulated by sprays with microalga biomass, such as the contents of total free amino acid in leaves, showing an interaction among treatments and cultivars. The AQ 25 stood out in both cultivars, promoting a remarkable 124.67% increase in amino acids in Alvará and 26.79% in Perfecta (Figure 5a).

Related to amino acid metabolism, the Alvará showed higher nitrate reductase enzyme activity in leaves (57.23%). Comparing treatments, AQ 40 promoted an increase of 40.63% in this enzyme activity in Alvará, over to the control. No changes were found in Perfecta (Figure 5c).

In the bulbs, comparing cultivars, Alvará showed amino acid content 112.92% higher than Perfecta at control, indicating better aptitude of this cultivar for amino acid synthesis and accumulation in bulbs. Comparing treatments, AQ25 promoted an increase of 18.39% in Alvará, while the treatments did not change the amino acid content in Perfecta bulbs (Figure 5d).

Comparing these results to those obtained at early bulbification stage in a pot experiment, a difference among cultivars appears regarding amino acid content and nitrate reductase activity. Perfecta presented the highest content of amino acids in leaves at early stage (Figure 2c), while Alvará at the harvest. Moreover, both cultivars had enzyme activity improved on average by AQ at early stage (Figure 2d), while only Alvará at the harvest (Figure 5c).

The outcome of the total sugars in the leaves at harvest show a 28.89% greater amount of sugars in the cultivar Perfecta compared to the Alvará. Comparing treatments, there were no significant differences among the control and the treatments AQ 5, AQ 25 and AQ 40 in Alvará, and among the control and AQ 25 and AQ 40 in Perfecta. On the other hand, AQ 15 caused a reduction in sugars in leaves of Alvará, as well as AQ 5 and AQ 15 cause reduction in Perfecta (Figure 5c).

It can be seen in the total sugar content of the bulbs that there was a higher concentration in Alvará, with 8.79% higher than Perfecta. The treatments, AQ 5, AQ 15, and AQ 40 promoted higher contents than the control in Alvará (46.05, 63.7 and 61.04%, respectively). In Perfecta, the treatments that presented the highest contents in relation to the control were AQ 5, AQ 15, and AQ 25 (45.3, 66.73 and 37.94%) (Figure 5d). These results indicate that AQ treatments stimulated the source-to-sink flow with sugar accumulation in bulbs, justifying the reduction in contents in leaves (Figure 5c).

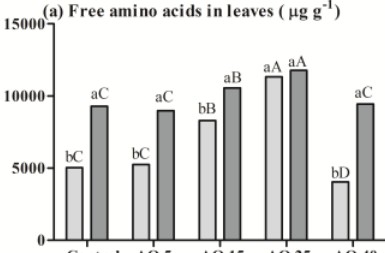
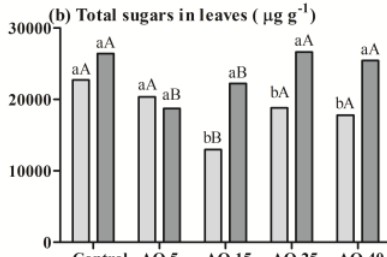
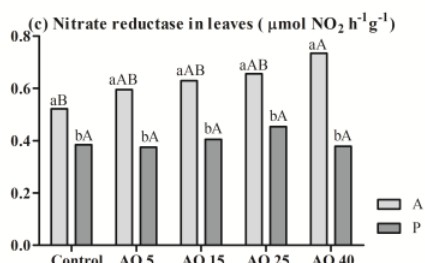

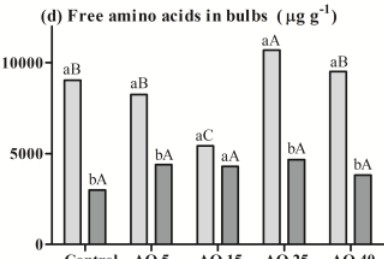
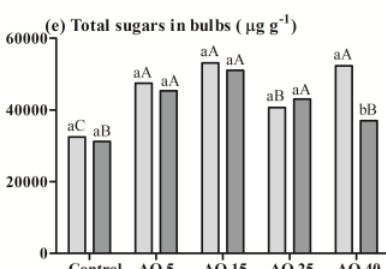

**Figure 5.** Free amino acids in leaves (**a**), total sugars in leaves (**b**), enzyme nitrate reductase activity in leaves (**c**), free amino acids in bulbs (**d**), total sugars in bulbs (**e**) in onion cultivars at 135 DAT submitted to foliar applications of biomass suspensions of the microalgae *A. quadricellulare* at different concentrations. Treatments: 0 g $L^{-1}$ (control); 0.05 g $L^{-1}$ (AQ 5); 0.15 g $L^{-1}$ (AQ 15); 0.25 g $L^{-1}$ (AQ 25); 0.4 g $L^{-1}$ (AQ 40). Lower-case letters in the column: cultivars; upper-case letters in the column: treatments. Means followed by the same letter in the same column do not differ statistically ($p < 0.05$) according to Tukey's test ($n = 4$).

## 4. Discussion

The productive performance of onions depends very much on the initial growth of plants right after transplanting [25]. The *Asterarcys quadricellulare* (CCAP 294/1) treatments provided greater initial plant growth, with increases in fresh mass of leaves, length and pseudostem diameter (Figure 1).

The increase in vegetable growth and biomass accumulation in the function of microalgae treatments was previously reported and related to increases in chlorophyll [7,12], in line with the rise in pigments in leaves sprayed with *A. quadricellulare*, particularly on AQ 25 in Alvará and AQ 5, AQ 15 and AQ 40 in Perfecta for chlorophyll; and on AQ 25 in the cultivar Alvará and AQ 5 and AQ 40 in Perfecta for carotenoids (Figure 2a,b).

The biomass of *Asterarcys quadricellulare* (CCAP 294/1) presents an L-free amino acids content of 90.94 mg $g^{-1}$. When absorbed by the plants, the free L-amino acids and peptides can be incorporated into metabolism according to the specific metabolic demands, such as for chlorophyll synthesis, or to stimulate physiological processes, therefore, acting as signaling molecules [8,26].

The microalga *Chlorella*, a Chlorophyta such as *A. quadricellulare*, also has a large number of L-amino acids in its composition, especially glutamic acid [27], which is considered to be a key amino acid in plant growth and development, as it is a precursor of other amino acids that are produced through transamination [28,29].

These possible metabolic actions of microalgae-free L-amino acid may justify the rise in the free amino acid content in onion leaves with AQ 15 sprays in Alvará and with AQ 5, AQ 15, and AQ 40 in Perfecta (Figure 2c) as an effect of *A. quadricellulare* stimulating the synthesis of amino acids in onion plants.

The role of *Asterarcys quadricellulare* (CCAP 294/1) in nitrogen assimilation can be also related to the activity of the nitrate reductase enzyme, where the AQ 25 and AQ 40 treatments showed improvement upon enzyme reaction (Figure 2d). In the assimilation of nitrate ($NO_3^-$), the nitrogen from the $NO_3^-$ is converted to nitrite ($NO_2$), a reaction catalyzed by the nitrate reductase enzyme. Subsequently, nitrite is converted into ammonia by the nitrite reductase enzyme and finally, the ammonium is converted into amino acids by the enzymes GS and GOGAT [30] and distributed over the plant.

In general terms, it is possible to associate the greater growth and mass accumulation of onion plants with the stimulus to the synthesis of amino acids and pigments promoted by some AQ treatments.

Plants treated with intermediate concentrations of *A. quadricellulare* present the highest mass accumulation in bulbs, demonstrating greater efficiency in the production and redistribution of photoassimilates and, consequently, in a larger bulb caliber (Figure 3).

A higher caliber was found with all AQ treatments in relation to the control in cultivar Perfecta and with AQ 15 and AQ 25 in cultivar Alvará (Figure 3). The larger calibers are related to the superior accumulation of mass of the bulbs, demonstrating that the microalgae *A. quadricellulare* promoted a greater redistribution of photoassimilates when compared to the control. The use of microalgae increasing the caliber and mass of red beet hypocotyls [7] and onion bulbs [4] also were previously reported, indicating an increase in the accumulation of photoassimilates in sink organs [31].

Around 80% of the dry mass of onion bulbs is carbohydrates mainly composed of reducing sugars glucose and fructose and the non-reducing sucrose and fructooligosaccharides. Sugars are the main products of photosynthesis and their transport from the source to the sink is very important for plant growth, being transported through phloem and accumulating in the sink, resulting in a greater gain in mass, size and yield [32].

The data (Figure 5) indicate that the treatments stimulated the source-to-sink flow with sugar accumulation in bulbs, justifying the reduction in contents in leaves at harvest, confirming the effectiveness of *A. quadricellulare* in yielding and redistributing photoassimilates from source to sink during the bulb filling stages.

These results explain the achievement with AQ, with the increase in yield owned by the accumulation of mass in bulbs and consequent caliber improvement. In Alvará, AQ25 promoted yield increase of 28.3%, while in Perfecta, the treatments AQ 5, AQ 15, and AQ 25 promoted yield increases by 26.36, 33.26, and 39.97%, respectively (Figure 4).

Products with plant-stimulant action (biostimulants and biofertilizers) may improve the total content of amino acid and soluble proteins in plants, contributing to better nitrogen assimilation and stimulation of amino acid metabolism [33].

Onion plants had higher levels of amino acids with the applications of *A. quadricellulare*. Amino acids form proteins and are precursors of molecules with fundamental biological roles in plants such as hormones, nucleotides, and cell wall polymers. They also have a role in the transport of nutrients in the plant, mainly nitrogen. They are largely involved in the plant's primary and secondary metabolism, leading to the synthesis of several compounds that influence production as well as the plant's tolerance to abiotic stresses [34,35].

The performance in nitrogen assimilation was evaluated through the activity of the nitrate reductase enzyme, which increased in the AQ treatments, particularly in the cultivar Alvará (Figure 5e).

The nitrate ($NO_3^-$) is reduced by nitrate reductase to nitrite ($NO_2^-$), and then imported into the plastid and reduced by nitrite reductase to ammonium ($NH_4^+$). The $NH_4^+$, whether taken directly from the environment or converted from $NO_3^-$, is assimilated by glutamine synthetase (GS) into glutamine, which provides N for virtually all cellular N-containing components directly or via glutamate [36]. Therefore, the increase in nitrate reductase synthesis may contribute to the N assimilation process.

The presence of free L-amino acid in microalgae is closely related to its plant growth promoting effect [7,37] associated to compounds with pathways derived from amino acids such as polyamines [6], besides glycosides and phytohormones [9]. The microalgae set of bioactive compounds can act as signaling molecules triggering multiple gene expression related to growth promotion, nutrient acquisitions, and adaptation to abiotic and biotic stresses [38].

In line with previous works using microalgae, through their free L-amino acid concentration and effects on the initial growth, yield, and metabolism stimulation with biochemical improvements, the microalga *A. quadricellulare* (CCAP 294/1) can be considered as a natural input, being a new alternative to growth of onion a nature-friendly way.

## 5. Conclusions

The use of *A. quadricellulare* (CCAP 294/1) promoted plant growth and increases in bulbs caliber and yield of onion-stimulating plant metabolism with increases in chlorophyll, carotenoids, amino acids, and nitrate reductase enzyme activity in leaves, also free amino acids and total sugar contents in bulbs, highlighting the biomass concentration of 0.25 g L$^{-1}$.

**Author Contributions:** Conceptualization, E.C.N.C., Á.F.M. and G.M.; methodology, E.C.N.C., Á.F.M.; software, J.d.O.A. and G.B.d.L.; validation, H.M.C.M., G.M. and Á.F.M.; formal analysis, E.C.N.C. and J.d.O.A.; investigation, E.C.N.C. and Á.F.M.; resources, Á.F.M. and G.M.; data curation, G.B.d.L. and H.M.C.M.; writing—original draft preparation, E.C.N.C., J.d.O.A., H.M.C.M. and G.B.d.L.; writing—review and editing, Á.F.M. and G.M.; visualization, E.C.N.C. and Á.F.M.; supervision, Á.F.M.; project administration, G.M.; funding acquisition, Á.F.M. and G.M. All authors have read and agreed to the published version of the manuscript.

**Funding:** This research was funded by Funpar project 03568.

**Institutional Review Board Statement:** Not applicable.

**Informed Consent Statement:** Not applicable.

**Data Availability Statement:** Not applicable.

**Acknowledgments:** To Coordenação de Aperfeiçoamento de Pessoal de Nível Superior-Brasil (CAPES)-Finance Code 001 providing Doctoral Scholarship to first author.

**Conflicts of Interest:** The authors declare no conflict of interest.

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
