# Peer review of "Microalga Biofertilizer Triggers Metabolic Changes Improving Onion Growth and Yield"

_horticulturae, doi:10.3390/horticulturae8030223_

Round 1
Reviewer 1 Report
Dear authors,
the paper "Microalga biofertilizer triggers metabolic changes improving onion growth and yield" submitted to horticulturae is an interesting study on a quite popular topic: the application of biofertilizer/biostimulant from natural origin in agriculture. However, experiments and trials with new substances on different plant species are needed to prove and indagate their effects and collect more and more information.
The manuscript is generally well written but I suggest the authors to check the use of English.
line 20: "in greenhouse" not "at";
lines 22-27: this sentence is too long;
Please check the space between the words (lines 22, 180, 195...);
line 75 and many others: the dot is not needed between g and L; between mL and plant (line 115); between t and ha (line 136); L and ha (line 146)....(line 167, 211, 212, 259, 302) please check the entire manuscript.
line 93: there are two "average" word in the same line, please check;
line 105: what do you mean with "suspended" on benches? I think you can find a more appropriate word;
line 156: do you mean "roles" or "rules";
line 208: please remove "F";
Please check the Figure caption in the entire manuscript and each link in the text. It is necessary to uniform the style. So decide if you put a dot after "Fig" or between the number and the letter (lines 195, 217, 221 and so on). Moreover, when you list the treatments in the caption, the AQ5 is always missing.;
line 229: please remove the ";" after 17.41;
Figure 3 b,c,d: can you insert the treatments name under the x axes;
Figure 4: you don't need to put "a" if there is only one graph. Moreover, I think it is better to style the graph as the Fig.3a, with the treatments on the x axes and the cv as label, is there any particular reason of this choice?
The discussion seems a little bit poor, can you improve this part and add a short conclusion of the entire work?
Author Response
Dear Reviewer,
The authors thanks for your important contribution to the improvement of manuscript quality.
All of your requests were addressed, included in attached file.

Reviewer 2 Report
Dear Authors,
Manuscript land-1613521, titled: ”Microalga biofertilizer triggers metabolic changes improving onion growth and yield” submitted to the journal Horticulture is scientific paper which is nice and clearly written. In paper to evaluate the effect of microalgae Asterarcys quadricellulare (CCAP 294/1) biomass as a source of bioactive free L-amino acids, on organic onion production, evaluating its action on metabolism, growth and yield of two cultivars and two cultivation methods.
Paper may be published after next minor corrections:
1). 57-63. line. Please delete paragraph from the introduction of 57-63. line and move it to Material and Method section.
- Thus, the following experiments were carried out: i) foliar sprays of A. quadricellulare spray-dried biomass on onion plants grown in pots in a greenhouse determining biochemical alterations and biometric variables of growth; ii) foliar sprays on organic field-grown onion plants; determining classification and yield of bulbs, as well as biochemical variables in leaves (chlorophyll, carotenoids, amino acids and the activity of the enzyme nitrate reductase) and in bulbs (free amino acids and total sugars).
- Please Highlight the conclusion in particular
- 535 line. Please in citate 20 separate the words in the sentence
- Lichtenthaler, H.K. Chlorophylls and carotenoids:Pigments ofphotosynthetic biomembranes. Methods in Enzymology. 535 1987, 148, 350-382. https://doi.org/10.1016/0076-6879(87)48036-1
Biostimulants and biofertilizers may improve the total content of amino acid and soluble proteins in Onion plants, contributing to better nitrogen assimilation and stimulation of amino acid metabolism. They are largely involved in the plant's primary and secondary metabolism, leading to the synthesis of compounds that influence increased production as well as the plant's tolerance to abiotic stresses. Therefore, this research has a great contribution to science.
Best regards,

Author Response
Dear Reviewer,
The authors thanks for your important contribution to the improvement of manuscript quality.
All of your requests were addressed, included in main document attached file.
We also thank you for your encouraging comments.
